# Hepatico-Duodenal Fistula Following Iatrogenic Strasberg Type E4 Bile Duct Injury: A Case Report

**DOI:** 10.3390/medicina59091621

**Published:** 2023-09-07

**Authors:** Bozhidar Hristov, Daniel Doykov, Desislav Stanchev, Krasimir Kraev, Petar Uchikov, Gancho Kostov, Siyana Valova, Eduard Tilkiyan, Katya Doykova, Mladen Doykov

**Affiliations:** 1Section “Gastroenterology”, Second Department of Internal Diseases, Medical Faculty, Medical University of Plovdiv, 6000 Plovdiv, Bulgaria; daniel_doykov@abv.bg (D.D.); dessislavs@gmail.com (D.S.); 2Gastroenterology Clinic, University Hospital “Kaspela”, 4001 Plovdiv, Bulgaria; 3Department of Propedeutics of Internal Diseases, Medical Faculty, Medical University of Plovdiv, 6000 Plovdiv, Bulgaria; kkraev@hotmail.com; 4Rheumatology Clinic, St. George University Hospital, 6000 Plovdiv, Bulgaria; 5Department of Special Surgery, Faculty of Medicine, Medical University of Plovdiv, 6000 Plovdiv, Bulgaria; puchikov@yahoo.com (P.U.); caspela@abv.bg (G.K.); 6Second Department of Surgery, St. George University Hospital, 4000 Plovdiv, Bulgaria; 7Department of Surgery, University Hospital “Kaspela”, 4001 Plovdiv, Bulgaria; 8Section “Nephrology”, Second Department of Internal Diseases, Medical Faculty, Medical University of Plovdiv, 6000 Plovdiv, Bulgaria; siyanavalova@abv.bg (S.V.); eet64@abv.bg (E.T.); 9Clinic of Nephrology, University Hospital “Kaspela”, 4001 Plovdiv, Bulgaria; 10Department of Diagnostic Imaging, Medical Faculty, Medical University of Plovdiv, 6000 Plovdiv, Bulgaria; katya.doykova@mu-plovdiv.bg; 11Department of Diagnostic Imaging, University Hospital “Kaspela”, 4001 Plovdiv, Bulgaria; 12Department of Urology and General Medicine, Medical Faculty, Medical University of Plovdiv, 6000 Plovdiv, Bulgaria; mladen.doykov@mu-plovdiv.bg; 13Clinic of Urology, University Hospital “Kaspela”, 4001 Plovdiv, Bulgaria

**Keywords:** laparoscopic, cholecystectomy, jatrogenic, bile ducts, fistula, hepaticoduodenal, stricture, endoscopic, dilation, stenting

## Abstract

Introduction: Gallstone disease (GSD) is among the most common disorders worldwide. Gallstones are established in up to 15% of the general population. Laparoscopic cholecystectomy (LC) has become the “gold standard” for treatment of GSD but is associated with a higher rate of certain complications, namely, bile duct injury (BDI). Biliary fistulas (BF) are a common presentation of BDI (44.1% of all patients); however, they are mainly external. Post-cholecystectomy internal BF are exceedingly rare. Case report: a 33-year Caucasian female was admitted with suspected BDI after LC. Strasberg type E4 BDI was established on endoscopic retrograde cholangiopancreatography (ERCP). Urgent laparotomy established biliary peritonitis. Delayed surgical reconstruction was planned and temporary external biliary drains were positioned in the right and left hepatic ducts. During follow-up, displacement of the drains occurred with subsequent evacuation of bile through the external fistula, which resolved spontaneously, without clinical and biochemical evidence of biliary obstruction or cholangitis. ERCP established bilio-duodenal fistula between the left hepatic duct (LHD) and duodenum, with a stricture at the level of the LHD. Endoscopic management was chosen with staged dilation and stenting of the fistulous tract over 18 months until fistula maturation and stricture resolution. One year after stent extraction, the patient remains symptom free. Discussion: Management of post-cholecystectomy BDI is challenging. The optimal approach is determined by the level and extent of ductal lesion defined according to different classifications (Strasberg, Bismuth, Hannover). Type E BDI are managed mainly surgically with a delayed surgical approach generally deemed preferable. Only three cases of choledocho-duodenal fistulas following LC BDI currently exist in the literature. Management is controversial, with expectant approach, surgical treatment (biliary reconstruction), or liver transplantation being described. Endoscopic treatment has not been described; however, in the current paper, it proved to be successful. More reports or larger case series are needed to confirm its applicability and effectiveness, especially in the long term.

## 1. Introduction

Gallstone disease (GSD) is among the most common disorders worldwide. Gallstones are established in up to 15% of the general population, with a symptomatic course seen in 11.7–32.8% of those cases [1]. Taking into consideration the incidence of the disease, understandably, laparoscopic cholecystectomy (LC) is among the most common surgical procedures. Surgical treatment is applied in 25.8% of asymptomatic patients and up to 50.7% of symptomatic ones, which translates to approximately 750,000 LC in the USA every year [2].

Since its introduction by Mouret in 1987, LC has become the “gold standard” for management of GSD. This fact at least partially contradicts the scientific evidence, which states that LC is more prone to certain complications compared to open surgery. Bile duct injury (BDI) incidence for instance is at least three times higher compared to the open approach, 0.6% vs. 0.2% [3,4]. Taking into consideration the absolute number of LC performed annually worldwide, the burden of this complication is not to be underestimated. Initially, BDI was attributed to the learning curve, but nation-wide studies in Japan, Denmark, Sweden and Great Britain, which assessed the complication rates of LC in the last 30 years, failed to establish a significant discrepancy between different time intervals. Single-port laparoscopic cholecystectomy is yet to confirm its safety profile, with initial studies suggesting an even higher complication rate compared to conventional LC [5]. In general, post-LC complications requiring subsequent interventional procedures or reoperation are associated with 8.8% short-term and 20% long-term mortality and a 0.8% need for a liver transplant [6]. Considering the frequency of the procedure, the absolute number of patients suffering such complications is not to be underestimated.

Biliary fistulas (BF) are a rare complication of GSD (1–2% of symptomatic patients) and represent an abnormal communication of the biliary system (most commonly, gall-bladder or common bile duct (CBD)) with adjacent structures [7]. They can connect the biliary system with the gastrointestinal tract (internal fistulas) or abdominal wall (external fistulas). Depending on their etiology, BF could be classified as primary or secondary. Primary fistulas are usually internal and occur as a result of persistent inflammation or neoplastic process. Secondary fistulas are iatrogenic in nature and typically develop after cholecystectomy either laparoscopic or open. Secondary biliary fistulas are the most common initial manifestation of BDI (established in 44.1% of cases) [8]. The vast majority of them are external (bilio-cutaneous), but occasionally, secondary internal (bilio-digestive) fistulas also could be established.

Herein, we present an unusual case of secondary bilio-digestive fistula formed between the left hepatic duct and duodenum, in a patient with iatrogenic BDI following LC. The clinical course as well as the diagnostic and therapeutic dilemmas are diligently described.

## 2. Case Report

### 2.1. Initial Diagnostic Evaluation

A 33-year-old Caucasian female was subjected to elective LC for symptomatic uncomplicated GSD. A conventional antegrade cholecystectomy was performed, with no adverse events noted intraoperatively. The postoperative period, however, was protracted with persistent albeit mild-to-moderate abdominal pain and low-grade fever up to 38.3 °C. There was no secretion from the drain positioned in the gall bladder bed. On the 3rd postoperative day, the patient’s complaints resolved partially, and it was decided to discharge her for outpatient observation. Upon removal of the drainage tube, however, spontaneous evacuation of substantial amount of bilious fluid occurred. At this point, bile leakage from the cystic duct was suspected and the patient was referred to our institution for further evaluation.

At admission, the patient’s general condition was preserved: blood pressure 110/60 mmHg, pulse rate 90/min, no fever. There was mild pain on palpation. Abdominal ultrasonography (US) was performed, which showed non-dilated bile ducts, fluid collection measuring 4/5 cm in the gall bladder bed, a small amount of fluid in the ileocecal region and above the urine bladder. Blood work established: leucocytes—13.3 × 10^9^ (3.5–10.5 × 10^9^/L), Hemoglobin (Hb)—152.0 g/L (120–140 g/L), thrombocytes—492 × 10^9^ (150–450 × 10^9^/L), Erythrocyte sedimentation rate (ESR)—22 mm/h (<15 mm/h), C-reactive protein (CRP)—17.5 mg/L (<5 mg/L), total bilirubin—59.12 µmol/L (7–22.2 µmol/L), conjugated bilirubin—33.2 µmol/L (1.2–7.2 µmol/L), alanine aminotransferase (ASAT)—82 U/L (<40 U/L), aspartate aminotransferase (ALAT)—160 U/L (<40 U/L), gamma-glutamyl transferase (GGT)—183.0 U/L (<52 U/L), alkaline phosphatase (AP)—325.0 U/L (<300 U/L). Urgent endoscopic retrograde cholangiopancreatography (ERCP) was scheduled the same day.

ERCP was performed under intravenous anesthesia using a combination of fentanyl, midazolam, and propofol. Antibiotic treatment (ceftriaxone 2.0 g i.v.) was initiated. The patient was placed in a supine position. A therapeutic duodenoscope Olympus TJF-160VR (Olympus, Hamburg, Germany) was introduced and placed at the second portion of the duodenum “en face” with the major duodenal papilla. Deep biliary cannulation was achieved using the standard guidewire technique (sphincterotome—TrueTome^TM^; Boston Scientific, Marlborough, MA, USA) and 0.035-inch guidewire (Jagwire^TM^ straight type, Boston Scientific, Marlborough, MA, USA). Cholangiography was obtained which showed non-dilated common bile duct with complete obstruction at the level of common hepatic duct (CHD) caused by a clip. Despite all efforts, advancement of the guidewire above the stenosis was impossible. It was concluded that the CHD was misidentified as the cystic duct (CD) and transection of the CHD above the clip was suspected (Strasberg type E4 BDI). Based on the ERCP, imaging, and clinical findings, biliary peritonitis following persistent bile leak was anticipated, so the patient was immediately transferred to the Surgical Department for further evaluation.

### 2.2. Surgical Approach

Upon laparotomy, biliary peritonitis was established. Inspection of the liver hilum revealed massive injury (probably thermal) of the CHD involving the biliary confluence (confirmed Strasberg type E4 BDI). The clip was found at the distal end of the CHD. Taking into consideration the severe inflammation in this region, immediate repair of the defect and biliary reconstruction was deemed to be hazardous. Instead, two percutaneous biliary catheters (10 fr) were inserted in the left and right hepatic duct to ensure drainage, lavage of the peritoneal cavity was performed, and finally, three abdominal drainage tubes were positioned in the small pelvic area and at the subhepatic space. Since second revision of the abdominal cavity was anticipated, laparostomy was constructed and the patient was transferred to an Intensive Care Unit for supportive care. Re-exploration of the abdominal cavity was performed four days later and found resolved biliary peritonitis with adequate function of the biliary drainage tubes. The patient’s condition improved substantially and she was discharged 3 days later. Readmission was planned in 4 weeks for further assessment and subsequent biliary reconstructive surgery.

Surgical intervention, however, was delayed by a COVID-19 infection, which was generally mild in nature and treated symptomatically in outpatient settings. Unfortunately, about a week prior to the planned surgery, displacement and spontaneous extraction of the positioned biliary drains occurred. Initially, spontaneous evacuation of bilious fluid through the already maturated external BF was noted. A few days later, though, bile flow seized and the patient noticed darkening of the urine, jaundice, and low-grade fever—37.6 °C. She was immediately admitted to the Department of surgery, where her lab tests showed elevation bilirubin to 82.5 µmol/L, ALAT—64 U/L, GGT—159 U/L, AP—358 U/L, CRP—242.9 mg/L, leucocyte—13.3 × 10^9^/L. Computer tomography (CT) was performed showing dilated intrahepatic bile ducts. Benign strictures of right and left HD were suspected and she was scheduled for surgery. Surprisingly, in the course of hospital admission, jaundice resolved and the patient even noticed normal coloration of the stools, which were persistently pale since the LC. A lab test also showed substantial improvement—bilirubin—33.2 µmol/L, ALAT—34 U/L, GGT—88 U/L, AP—228 U/L, CRP—15.6 mg/L, leucocyte—10.1 × 10^9^/L. At this point, the patient was again referred to Gastroenterology Department for evaluation.

### 2.3. Endoscopic Management

ERCP was performed in line with the described protocol. Upon transpapillary cannulation, no dynamic in the fluoroscopic finding was noted—non-dilated CBD, with proximal obstruction caused by a clip at the level of CHD. Careful withdrawal of the duodenoscope, however, revealed a fistulous opening at the level of bulbus duodeni with a flow of bile. Bilio-digestive fistula was naturally suspected. The fistulous opening was cannulated using a sphincterotome and a 0.025-inch straight guidewire (Figure 1). A short stricture was noted about 10 mm proximal to the fistulous opening, which was initially hard to negotiate through, but eventually, deep biliary cannulation was achieved. Fluoroscopy showed then a biliodigestive fistula formed between duodenum and the left hepatic duct (LHD), with a high-grade stenosis at the distal end of the duct. Further opacification of the biliary tree revealed an anatomical variation of the bile ducts with the right posterior duct draining into the left hepatic duct (Type 3A according to the intrahepatic duct anatomy classification; see Appendix A) (Figure 2). Considering this fact as well as the lack of clinical signs of cholangitis, it was decided that stenting of the left hepatic duct would ensure adequate biliary drainage and might preclude the need for surgery. The stricture was dilated to 8 mm using a biliary balloon dilation catheter (Hurricane Rx, Boston Scientific, Marlborough, MA, USA) and two straight plastic stents were inserted (Endoflex, Hamburg, Germany)—7 fr/6 cm in the 2nd segment branch and 10 fr/6 cm 3rd segment branch (Figure 3). There were no adverse events (AE) defined according to the European Society for Gastrointestinal Endoscopy (ESGE) and American Society for Gastrointestinal Endoscopy (ASGE) guidelines [9,10]. The patient was discharged 3 days later, with readmission for reevaluation planned six months later.

At six months, the patient was readmitted and ERCP performed electively. Fluoroscopy revealed significant, though incomplete resolution of the stricture at the level of LHD. Further balloon dilation to 10 mm was performed, with subsequent insertion of 3 straight plastic stents—10 fr/6 cm in the 2nd segment branch, 10 fr/6 cm and 8.5 fr/6 cm in the 3rd segment branch. No ERCP-related adverse events were noted.

The patient remained symptom-free for the next 12 months. At one year, elective ERCP was performed. Complete resolution of the stricture was found, with one of the stents proximally migrated in the HD (discussed as an AE in ASGE but not ESGE guidelines). All biliary prostheses were removed with a snare, with the migrated stent being extracted using a standard upper endoscope, which was easily introduced through the fistula.

After one year of follow-up, the woman remains symptom free. Regular blood tests and abdominal US show no signs of biliary obstruction.

## 3. Results

Since its introduction in 1987, LC has been established as the “gold standard” in management of GSD. While undeniably advantageous in terms of extent of surgical trauma, recovery time and cost-effectiveness, LC has been persistently associated with a higher rate of specific complications, namely, BDI (0.5–1.4% vs. 0.08–0.3% for open cholecystectomy) [11,12]. Despite the completion of the learning curve, studies conducted from the 1980s to 2015 have failed to establish any significant drop in BDI incidence [13]. Adequate management of BDI is of paramount importance considering the fact that major injuries result in 40–50% short-term morbidity and 2–4% mortality [13,14,15,16,17]. Late complications, including biliary strictures, anastomotic strictures, recurrent cholangitis, and biliary cirrhosis, increase the burden for the patient even more.

Management strategy is largely determined on the type of BDI and the time of recognition. Various classifications of BDI exist. The most commonly utilized ones are presented in Table 1 and Table 2 and Figure 1.

Strasberg classification [19] allows for differentiation between minor (bile leakage from the cystic duct or aberrant right sectoral branch) and major BDI. Type E of the Strasberg classification is an analogue of the Bismuth classification. The Strasberg classification can be easily applied in clinical practice with one major disadvantage of not accounting for the presence of vascular involvement. In the field of interventional endoscopy, it is the most extensively utilized model of BDI.

First proposed by Bektas et al. [20], Hannover classification (Figure 4) overcomes the chief disadvantage of Strasberg classification by describing the presence of vascular injuries in groups C and D.

In an attempt to comprise all existing classifications into a composite, the ATOM (Anatomic Time Of detection Mechanism) classification was developed [21]. It provides an exhaustive definition of BDI based on the location, presence of concomitant vascular injury, time of recognition, and mechanism of damage. It covers all possible clinical scenarios, which would undoubtedly improve subsequent decision making. Its complexity, however, narrows its application in routine practice. Of note, reporting of injuries evaluated through ERCP in accordance with ATOM classification would also be impossible.

In terms of timing, BDI could be identified either intraoperatively or in the early post-operative period. The ideal scenario would be intraoperative recognition and immediate repair of the lesion. Unfortunately, such is seen in merely 15–30% (23% on average) of the cases [22]. Additionally, in the absence of a surgeon with hepato-biliary expertise, the success rate of primary repair is considered to be low, with Stewart and Way claiming that only 13% of repairs performed by the index surgeon without HPB expertise were successful [23]. In our case, the BDI was not identified during the LC, which is in line with the presented data.

In the post-operative period, typical presentation would include bilious effusion from the drains or laparoscopic ports, diffuse abdominal pain, nausea, fever, impaired intestinal motility, bile collections, signs of peritonitis, leukocytosis, mixed hyperbilirubinemia. An obstructive pattern in liver function tests accompanied by jaundice is frequent in the biliary obstruction scenario. If not identified during the first postoperative week, patients have an insidious evolution with relapsing abdominal pain and cholangitis. Jaundice is not always present immediately after bile duct injury. Some partial stenosis and isolated sectorial right duct lesions (Strasberg B and C) present with abdominal pain, pruritus, general weakness, fever, and intermittent alterations of liver function tests. The late clinical course of bile duct injury leads to chronic liver disease, cirrhosis, and portal hypertension.

In the current case, the clinical course was typical with low-grade fever, diffuse abdominal pain, and an increase of liver enzymes and inflammatory biomarkers. There was, however, no bilious effusion from the drains, which led to a considerable diagnostic delay. It is our opinion this occurrence was due to an inadequate positioning or overlooked obstruction of the drainage tubes.

Once suspected, the diagnostic approach in BDI is focused on establishing the nature of the lesion and determining the optimal therapeutic option. Currently, the optimal diagnostic modality is considered to be magnetic resonance cholangiopancreatography (MRCP) with an estimated sensitivity (Se) of 91–95% and specificity (Sp) of 98%. Alternative methods include ERCP (Se—80–93%; Sp—100%); Percutaneous transhepatic cholangiography (PTC) (Se—95.8%; Sp—81.2%), CT (Se—78%; Sp—86%), and US (Se—20–90%; Sp—62–100%).

In our case, MRCP was not performed since it was not readily available and the need for therapeutic procedure was anticipated to be high, justifying the early transition to invasive procedures (ERCP). Abdominal ultrasonography established the presence of biliary obstruction and fluid collections in the abdomen, but failed to determine the nature of the lesion, which is to be expected. Eventually, a type E4 BDI (as per Strasberg classification) was established on ERCP, which was reconfirmed during the subsequent surgical intervention. Arguably, performing an MRCP would obviate the need of ERCP in this case since endoscopic treatment of type E4 BDI is impossible. On the other hand, urgently performed ERCP ensured an accurate diagnosis and eliminated any further diagnostic delay, which we consider to be crucial in the setting of biliary peritonitis.

Surgical management included three important aspects—laparotomy, avoidance of immediate reconstruction (only biliary drainage), and temporary abdominal closure (TAC) (laparostomy). The rationale for an open surgical approach is understandable, taking into consideration the anticipated severe inflammatory changes in the liver hilum and biliary peritonitis. The question of early or delayed surgical repair of BDI is still debatable. Many authors consider the delayed repair to be associated with fewer complications in the short and long term [24,25]. Arguments to support this are the opportunity to stabilize patients’ condition, control sepsis, improve focal inflammation, and microvascular damage to the bile ducts, all of which interfere negatively on the clinical outcomes. A staged approach with definitive reconstruction planned about 45 days after the primary procedure is largely adopted at our institution. The decision to leave an “Open abdomen” (OA) also could be qualified as debatable. Numerous studies are yet to establish the optimal indications for OA [26]. In our case, the decision was derived by the presence of severe biliary peritonitis and the anticipated need for a “second look” operation to confirm control over abdominal sepsis and optimal position of biliary drains. Due to the lack of clear guidelines, such a decision is largely made on a case-by-case basis and at the discretion of the surgeon.

The current case presents an interesting evolution of a major BDI into bilio-digestive (hepatico-duodenal) fistula. Upon the literature review, we identified three cases of choledocho-duodenal fistulas following post-cholecystectomy BDI [27,28,29]. All of them follow a similar clinical scenario, which was also observed in our case. Upon presentation of bile leakage, laparotomy is performed to diagnose type-E BDI. In all cases, primary repair was not attempted but temporary external drainage was ensured by means of biliary drains (in two cases) or drains in the subhepatic space. In all cases the formation of biliodigestive fistula was marked by cessation of bile flow from the drainage tubes. In the Kakaei et al. [27] report, displacement of the drainage tubes also occurred, which was seen in our case as well. While the clinical presentation was largely identical, the management strategy was different in all cases. Kakaei et al. adopted an expectant approach with the intent of subsequent reconstructive surgery, but such proved to be impossible and the patient was referred for liver transplantation. A conservative approach was preferred by Yilmaz et al., but since the fistula stricture with subsequent cholangitis developed during follow-up, surgical repair was performed eventually. On the other hand, Gallagher et al. decided for immediate surgical repair (Roux-en-Y hepaticojejunostomy) anticipating a high probability of fistula stenosis and subsequent complications.

To our knowledge, this is the first case to report endoscopic management of spontaneous hepatico-duodenal fistula, following iatrogenic BDI. Several facts were taken under consideration. Firstly, since the diagnosis was set in the course of ERCP and up on successful cannulation of the intrahepatic bile ducts, it was decided that management of the stricture as any other benign biliary stricture is a viable option. Additionally, though high-grade, the stenotic region was quite short, which was considered favorable. Upon cholangiography, the fistulous tract was established to be between the left hepatic duct and duodenum. This would preclude endoscopic treatment, but as already mentioned, further fluoroscopic evaluation found a variation of biliary anatomy with the right posterior duct draining into the left hepatic duct (Type 3A according to the intrahepatic duct anatomy classification; see Appendix A). The lack of cholangitis and the opportunity to drain at least six liver segments avoiding major surgical procedure drove our decision to attempt endoscopic therapy. The patient’s preferences were also accounted for.

This is the first case to report fistula formation between the left hepatic duct and duodenum. In all three reported cases, so far, the communication was distal to the biliary confluence.

The therapeutic strategy was identical to the approach to any benign biliary stricture with staged dilations and gradual increase of stent size and number. The period of stenting was prolonged to 18 months in total, in an attempt to achieve complete stricture resolution and fistula maturation. There are no existing reports of endoscopic management for this indication; hence, comparison with alternative techniques is impossible. Though quite long (1 year), probably a longer follow-up is justified to confirm the definitive stricture resolution.

## 4. Conclusions

Bilio-duodenal fistulas following post-cholecystectomy BDI are exceedingly rare. Delayed biliary reconstruction by means of Roux-en-Y hepaticojejunostomy is the most extensively utilized technique. Current case suggests that endoscopic treatment is a viable option in selected patients which ensures excellent short- and mid-term results obviating the need for major surgery. Future studies are needed to prove its applicability and especially its effectiveness in the long term.

## Figures and Tables

**Figure 1 medicina-59-01621-f001:**
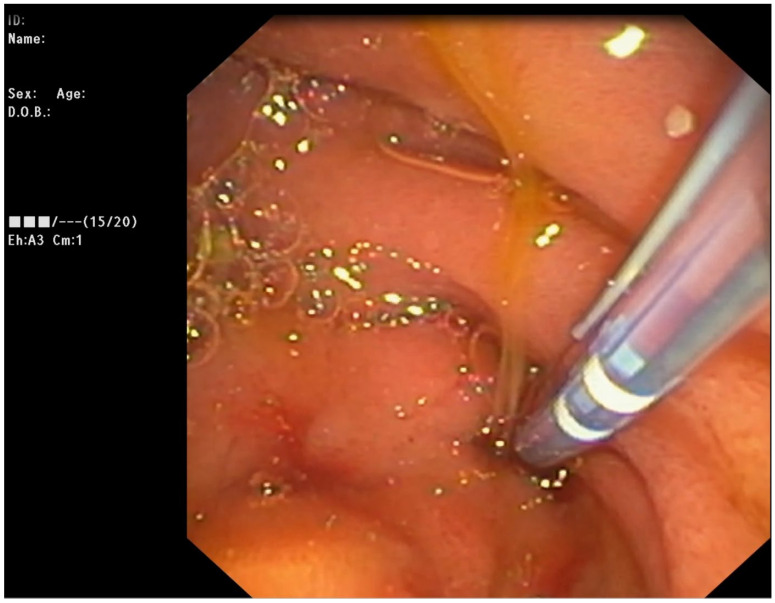
Transduodenal endoscopic cannulation through the distal orifice of the hepatico-duodenal fistula.

**Figure 2 medicina-59-01621-f002:**
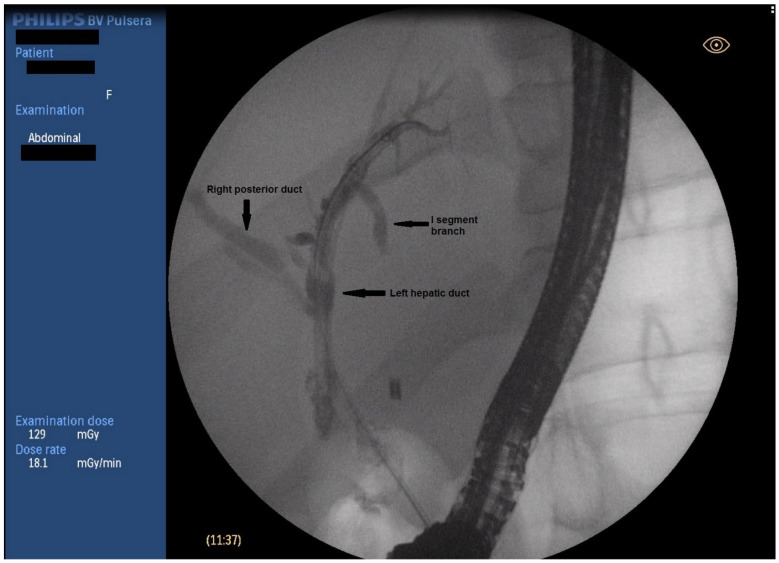
Fluoroscopic image of biliary anatomy.

**Figure 3 medicina-59-01621-f003:**
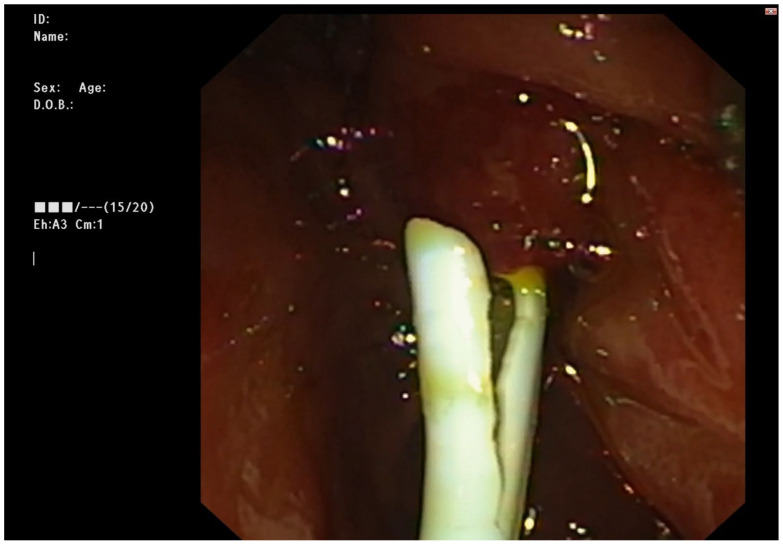
1st stent inserted through the fistula with bile flowing.

**Figure 4 medicina-59-01621-f004:**
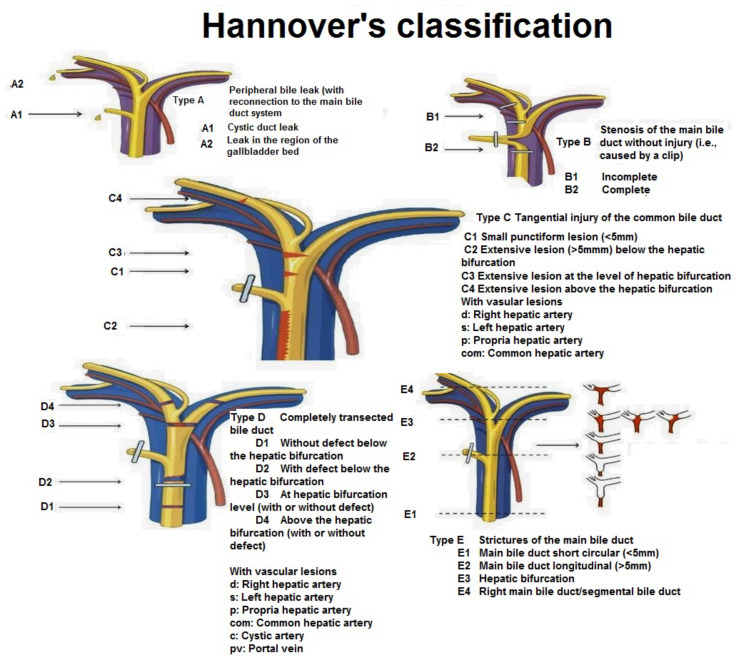
Hannover classification (Source: https://slideplayer.com/slide/12843812/, accessed on 9 August 2023).

**Table 1 medicina-59-01621-t001:** Bismuth–Corlette classification [18].

Type	Description of BDI
I	Involves the common bile duct and low common hepatic duct (CHD) > 2 cm from the hepatic duct confluence.
II	Involves the proximal CHD < 2 cm from the confluence.
III	Hilar injury with CHD confluence intact.
IV	Destruction of the confluence when the right and left hepatic ducts become separate.
V	Aberrant right posterior hepatic duct injury with or without concomitant injury of CHD.

**Table 2 medicina-59-01621-t002:** Strasberg classification.

Type	Description of BDI
A	Bile leak from cystic duct stump or minor biliary radical in gallbladder fossa.
B	Occluded right posterior sectoral duct.
C	Bile leak from divided right posterior sectoral duct.
D	Bile leak from main bile duct without major tissue loss.
E1	Transected main bile duct with a stricture more than 2 cm from the hilus
E2	Transected main bile duct with a stricture less than 2 cm from the hilus
E3	Stricture of the hilus with right and left ducts in communication
E4	Stricture of the hilus with separation of right and left ducts.
E5	Stricture of the main bile duct and the right posterior sectoral duct.

## Data Availability

Data available on request due to ethical restrictions.

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
