# Peer review of "Hepatico-Duodenal Fistula Following Iatrogenic Strasberg Type E4 Bile Duct Injury: A Case Report"

_medicina, 2023, doi:10.3390/medicina59091621_

Round 1

Reviewer 1 Report

Dear Authors,

Thank you for submitting your manuscript. Here are some comments for your consideration:

    •  
  1. Originality/Novelty:

    • The focus on bilio-duodenal fistulas following post-cholecystectomy BDI offers an important perspective on the management of rare complication but with serious consequences.
  2. Significance of Content:

    • The content is significant, especially for clinicians dealing with post-operative complications.
  3. Quality of Presentation:

    • The Results section is coherent, but consider adding sub-headings for clarity.
    • Please review figure numbering for Hannover's classification. Also, the figure quality is poor, image is distorted and text is not readable. Consider adding the source of the picture.
  4. Interest to Readers:

    • The content is relevant, especially for surgeons and gastroenterologists.
  5. Methods:

    • Methods seem to be accurately described.
    • Ensure all techniques and procedures are described in sufficient detail.
  6. Introduction:

    • Provides good background. Consider emphasizing the prevalence and significance of BDI to stress the importance.
  7. Conclusions:

    • Conclusions appear to align with the results.
    • References need to be adapted to MDPI style.
  1. Quality of English Language:

    • Overall, the manuscript is written in clear English.
    • Some sections may benefit from further proofreading to ensure grammar consistency. Consider the help of a native speaker. For example: "gallstones are established" , "golds standard"

Author Response

Response to Reviewer 1 Comments

Thank you very much for reviewing our manuscript. I find your remarks to be well-founded and largely justified and increasing the scientific value of the article. We tried to address all recommendations in the revised manuscript and corrections were applied where required.

Reviewer 2 Report

1) I propose to change the "key words": laparoscopic, cholecystectomy, jatrogenic,bile ducts, fistula, hepaticoduodenal, stricture, endoscopic, dilation, stenting; In the introduction absense references 1 and 2.

2) I did not understood the indications to laparotomy, laparostomy and reexploration:

a) small fluid collections were detected by US in the gallbladder bed, ileocaecal region and small pelvis without fibrin and level of leucocites was 13,3; ESR 22; CRP 15.5. It is non «severe biliary peritonitis». CT could confirm US data. « Massive ingury» in the liver hilum area had «thermal» etiology. It is biliary peritonitis no too;

b) WSES 2020 (World society of emergency surgery) in major injury (Strasberg E) diagnosed within 72 h after LC recommend Roux-en-Y hepaticojejunostomy (HJS) for experienced surgical centres and till 72 h to 3 weeks – abdominal cavity lavage as a first step to HJS;

с) EAES ( European Association for Endoscopic Surgery) recommend laparoscopy, lavage and draining of abdominal cavity in these cases ( Guidelines 2014). Authors been able to perform percutaneous biliary tree drainage during laparoscopy;

d) Which reason of by leakage authors seen intraoperatively?;

e) I propose to change the title of figure 1: «Transduodenal endoscopic cannulation through the distal orifice of the hepatico-duodenal fistula»;

f) Authors non describing at the pages 7-8 a famous ATOM-classification proposed EAES in 2013;

g) For my opinion « stenotic region» between left hepatic s wall and duodenal wall consisted from cicatricial postinflammed tissues without endothelium or duodenal epithelium covering above latters. They are may be transformed to stricture in long-term, unfortunately;

h) 48,3 % of References were published before 2015.

My remarks to authors are non principle. My congratulations with first endoscopic treatment of the rare jatrogenic hepatico-duodenal fistula and good short – and mid-term results.
